# Synthesis and Characteristics of Eco-Friendly 3D Printing Material Based on Waterborne Polyurethane

**DOI:** 10.3390/polym13010044

**Published:** 2020-12-24

**Authors:** Ji-Hong Bae, Jong Chan Won, Won Bin Lim, Jin Gyu Min, Ju Hong Lee, Chung Ryeol Kwon, Gyu Hyeok Lee, Pilho Huh

**Affiliations:** Department of Polymer Science and Engineering, Pusan National University, Busan 609-735, Korea; jihong.bae@pusan.ac.kr (J.-H.B.); jcwon@pusan.ac.kr (J.C.W.); freewonbin@nate.com (W.B.L.); jg_min0629@naver.com (J.G.M.); dlwnghd15@pusan.ac.kr (J.H.L.); cndfuf0901@pusan.ac.kr (C.R.K.); ghl8314@pusan.ac.kr (G.H.L.)

**Keywords:** waterborne polyurethane, photocurable resin, photopolymer, three-dimensional printing architectures, digital light processing

## Abstract

Photo-cured 3D architectures are successfully printed using the designed waterborne polyurethane-acrylate (WPUA) formulation. A WPUA series is synthesized in the presence of polycaprolactone diol (PCL) and 4,4′-methylene dicyclohexyl diisocyanate (H_12_MDI) as the soft segment part, dimethylolbutanoic acid (DMBA) as the emulsifier, and triethylamine (TEA) as the neutralizer, as a function of prepolymer molecular weight. The compatibility of WPUA and the photo-activating acryl monomer is as a key factor to guarantee the high resolution of 3D digital light processing (DLP) printing. The optimized blending formulations are tuned by using triacrylate monomers instead of diacrylate derivatives. For the high-accuracy and fine features of 3D DLP printing, WPUA are designed to be a suitable molecular structure for a 385 nm wavelength source, and the target viscosity is achieved in the range from 150 to 250 Cp. Photo-cured 3D architectures based on WPUA exhibit good flexural strength and high resolution.

## 1. Introduction

3D printing is regarded as a technology that will revolutionize the future manufacturing industry and has transformative potential that can change all production methods [1,2,3]. In medical industries, 3D printing has been expected to be one of the most innovative tools in the related technologies [4,5]. 3D printing is being advanced to provide a variety of customized implants, biodegradable artificial supports, and even artificial organs in the future [6]. Photo-curable resins can be effectively used as 3D materials based on various printer types such as SLA (Stereo Lithography Apparatus), DLP (Digital Light Processing), and Polyjet [7,8,9,10,11]. A soft active resin series can make it mainly suitable for high-accuracy or -resolution and user-customized strength or bending properties. The DLP method has been extensively used as a 3D printing method, which provides fast output speed and consistently high accuracy. However, there is a need to supplement material limitations and spatially controlled solidification. Waterborne polyurethanes (WPUs) have been one of few product types that can deliver results in both areas of the eco-environment and eco-safety because of non-pollution of the air or the production of non-toxic wastewater [12,13,14,15]. In addition to offering the high-performance modern applications needed, WPUs can contain no residual-free isocyanate and a low odor, making them safer and easier to use than solvent-borne polyurethanes. These WPU systems may be formulated as graphic art inks, adhesives for shoes and textiles, and can be used as coatings on flexible and rigid substrates [16]. Up to recently, few efforts have been carried out to develop fully 3D printed architectures of photo-curable WPU resin that are environmentally friendly, volatile organic compounds (VOCs)-free, and have lower hazardous air pollutants (HAPs).

In this study, we report a blending formulation series of the WPU matrix and photo-active triacrylate derivatives that is suitable for a UV-curing system based on 3D DLP printing. The resolution and mechanical properties of the DLP-printed waterborne polyurethane-acrylate (WPUA) structure are easily controlled by mixing a monofunctional monomer of urethane acrylate and a trifunctional cross-linker of triacrylate. The WPUA system prepared in this work will significantly apply to bendable and flexible 3D architectures and devices such as flexible electronics, soft robots, and many other fields.

## 2. Material and Methods

Scheme 1 presents a schematic diagram of the basic procedure to synthesize a WPU series. Polycaprolactone diol (PCL, Mn = 530 g/mol, Merck KGaA, Darmstadt, Germany) as a polyester polyol, 4,4’-methylene dicyclohexyl diisocyanate (H_12_MDI, Tokyo Chemical Industry Co., Ltd., Tokyo, Japan) as an isocyanate, and dimethylolbutanoic acid (DMBA, Merck KGaA, Darmstadt, Germany) as an internal emulsifier were dried over 6 h in a vacuum prior to use. Acetone from Duksan Chemical Co., Ltd. (Seoul, Korea) was used as the solvent to prepare the WPUs. At a total content of 30 g, the amount of DMBA inserted for the ionic group was 5 wt.%. In the first step of the synthetic procedure, PCL and H_12_MDI were stirred mechanically to form the prepolymer under a stirring speed of 200 rpm and N_2_ purging for 7 h at 80 °C. Then, 1.5 g of DMBA (0.01 mol) and 2.6561 g of H_12_MDI (0.01 mol) in 5 mL acetone were vigorously stirred to prepare the soft/hard backbone structure of polyurethane (PU) at 80 °C for 1 h. After cooling to 50 °C, 1.02447 g of triethylamine (TEA, Merck KGaA, Darmstadt, Germany) was sequentially added at the same molar ratio (0.01 mol) of DMBA. The resultant PU was dispersed in 70 mL DI water under a stirring speed of 600 rpm at 30 °C. To remove the residual acetone solvent, WPU was heated for 30 min at 60 °C. Table 1 presents the split composition of the comparative groups in molar ratio.

## 3. Results

The basic mechanical properties of a WPUA series for printing quality in 3D DLP are evaluated based on a function of molecular weight (MW). Figure 1 presents the mechanical behavior and particle size of the WPU system with different MW ranging from 3000 to 30,000. With increasing MW, the tensile strengths of the WPU series increase in strength values from 37 to 73 MPa. In this WPU system, the micelle structure increases with increasing in the MW. The presence of hydrogen bonds (H-bonds) between PCL/H_12_MDI consisting of soft domains is attributed to higher strength and bigger particle size. Upon mechanical loading, the strong intensity by H-bonds strengthens the friction-resistance and therefore, results in the high strength of WPU. By changing the MW of WPU, the basic mechanical performance of the 3D printable WPU can be readily tuned for the desired applications.

The photo-curable formulation of WPUA with enhanced compatibility is evaluated by various ratios of WPU/diacrylate and WPU/triacrylate ranging from 0.2 to 1.0. Figure 2 shows the miscible and immiscible phase images of WPU/multi-acrylate to confirm the dispersibility of the acrylate derivatives with WPU. Compared to WPU/diacrylate, triacrylates are well dispersed in the continuous phase of the WPU matrix, as shown in Figure 2c,d. This result could be due to the relative strengthening of intermolecular H-bonds in the arrangement of the triacrylates and polymer chains of WPU.

The viscosity of the WPUA resin can be one of the key parameters to guarantee the resolution and processability of 3D DLP printing. Figure 3 presents the viscosity testing results of the modified WPUA series by tuning WPU at 0, 5, 10, 20, and 30 wt.%. The viscosity is dependent on WPU content in the range 60 to 300 mPa·s. The high addition of WPU sharply increases the viscosity of the modified WPUA series, but the viscosity of 0, 5, and 10 wt.% WPU is low enough for 3D DLP printing and the viscosity of 30 wt.% WPU is high enough. WPUA by tuning 20 wt.% WPU is an acceptable viscosity, which can be suitable for 3D DLP printing.

The material properties of the WPUA series are evaluated to analyze the effect of tuning WPU contents on the bending property as well as flow behavior. Figure 4 shows the flexural behaviors of the modified WPUA series with various WPU percentages ranging from 0 to 30 wt.%. The increase in the weight fraction of WPU leads to the enhanced bending ability of the WPUA series but also results in the decrease in hardness, as shown in Figure 4. The increase in WPU from 0 to 30 wt.% improves the flexural strength from 12 to 25 MPa. The result indicates that the change of WPU content can be a simple approach to tune the flexural and hardness strength of the commercially available UV-curable WPUA by changing WPU contents.

The DLP printing process can have some limitations that can interact, such as bad adhesion and deformations between each layer. There are also internal defects like cracks or a porous appearance in printed pieces. The high-accuracy shoe- and line-like architectures by 3D DLP printing are displayed in Figure 5. The photographs of the complex architectures in Figure 5 reveal a high feature accuracy and excellent surface finishing. The DLP printing process of the WPUA resin progresses well through a build-speed of 10 μm/s in a layer-by-layer approach. The 3D printed shoe- and line-like samples correspond well to both the complex digital designs, with respect to dimensional accuracy and surface quality. The results also suggest that WPUA resin can be considered as a potential UV-curable material to fabricate complex 3D macro-architectures.

Fine resolution can provide a great influence on the printing quality and functionality of a 3D object. Figure 6 presents the SEM images of microscale layers to confirm the presence of minimal defects in the micro- or submicro-surfaces. The layer-by-layer is uniformly printed with thickness of 100 μm, as shown in Figure 6a. The edges of the internal layers in Figure 6b exhibit smooth surfaces with high-resolution patterns. The results indicate that good-quality objects with a high-resolution as low as 5 μm can successfully be printed using a WPUA series. From the SEM images in Figure 6c–e, the amount of cracks or porous appearance in the printed piece increases with the increase in WPU contents and also results in the increase in roughness. These defects may be addressed by controlling the printing conditions such as modification of the WPUA formulation, temperature, and UV-exposure time.

## 4. Conclusions

In summary, we report novel UV-curable formulations of WPU-based resin for 3D DLP printing. The 20 wt.% WPU-based resin provided a 3D elastic object that could reach a hardness and flexural strength of up to 95 and 23.0 MPa, respectively, which were higher than those of the normal UV-curable elastomers. The UV-curable WPUA series by changing WPU contents content can be a useful approach to tune the flexural and hardness strength of the commercially available 3D DLP-printed objects. The photo-curable WPUA system can also be applied as a promising candidate to fabricate bendable and flexible 3D architectures and devices by 3D DLP printing.

## Data Availability

Not applicable.

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
