# Peer review of "Synthesis and Characteristics of Eco-Friendly 3D Printing Material Based on Waterborne Polyurethane"

_polymers, 2020, doi:10.3390/polym13010044_

Round 1
Reviewer 1 Report
This manuscript reported the fabrication of photo-cured 3D printing architectures by appropriately designing waterborne polyurethane-acrylate (WPUA), which may be applied as an eco-friendly material in 3D printing fields. Besides, by using triacrylate monomer instead of diacrylate can obtain a complete blend compound. What is commendable is that, direct application of a 20 wt.% WPU-based formulation in normal DLP printing was proven to be the successful formation of a complex 3D architecture with good hardness and excellent flexural strength, which may have a guiding role in the design of this type of materials. However, there are still some minor questions for the authors to take into consideration while revising this work.
- Some format errors in part of “Material and Methods” should be corrected, for example, “70ml” in line 54 should be changed into “70 mL”, the molecular weights format shout be changed into (Mn) in line 56 and “g/mol” should be added after the number. Besides the spaces should be added between the numbers and units in the article.
- There are minor grammatical errors in the article. Such as the statement “too low a viscosity” in line 87 should be changed into “a too low viscosity”.
- In Figure 7, the surface of the hole in the material of Monomer + WPU 20% (b) seems rougher than (a) and (b), What is the possible reason of this phenomenon?
Author Response
(#1, #2) Based on the comments you gave me, I have revised the paper as a whole.
(#3) The amount of crack or porous appearance in the printed piece increases with the increase of WPU contents and also results in the increase of roughness.
These defects may be addressed by controlling the printing conditions such as modification of WPUA formulation, temperature, and UV-exposure time.

Reviewer 2 Report
Bae et al. present in this manuscript the synthesis and characteristics of a 3D printing material based on waterborne polyurethane, prepared from polycaprolactone diol, 4,4’-methylene dicyclohexyl diisocyanate, dimethylolbutanoic acid, and triethylamine. The topic of the manuscript is interesting and fits the scope of the journal Polymers. Unfortunately, the manuscript is not carefully written and products have not been structurally identified.
Comments:
--- line 11: Please correct “trimethylamine” to “triethylamine”. Similarly in line 53.
--- Scheme 1: The formula of polycaprolactone diol in Scheme 1 is wrong (the formula shown is not a diol). Please check (or take from the home page of the producer, Sigma-Aldrich). Consequently, the formula of OCN--------NCO is wrong. The formula of DMBA is wrong: the chain terminating group (top) is CH3 not CH2.
--- Lines 51-55: A more precise recipe is required; amounts of reagents and solvent have to be provided. (E.g.: There is no meaning to mention that the mixture was dispersed in 70 ml water if the amount of the mixture is not known.)
--- Materials and Methods: The final polymer synthesis, with materials and precise description, has to be provided here (not only the prepolymer synthesis). It is hidden in the text.
--- The composition and purity of the final polymer have not been investigated in this work, however, it is a requirement. Authors should provide proofs that they synthesized the desired materials.
--- Line 106: “100um” should be “100 μm”
--- Line 107: “(Figure 4(a))” should be (Figure 5(a)).
--- References: please provide references as it is required by the journal; Author 1, A.B.; Author 2, C.D. Title of the article. Abbreviated Journal Name Year, Volume, page range.
Author Response
Based on the your comments, I have revised the paper as a whole.
